# *Cryptosporidium* Prevalence in Calves and Geese Co-Grazing on Four Livestock Farms Surrounding Two Reservoirs Supplying Public Water to Mainland Orkney, Scotland

**DOI:** 10.3390/microorganisms7110513

**Published:** 2019-10-30

**Authors:** Beth Wells, Claire Paton, Ross Bacchetti, Hannah Shaw, William Stewart, James Plowman, Frank Katzer, Elisabeth A Innes

**Affiliations:** 1Moredun Research Institute, Pentlands Science Park, Penicuik, Midlothian EH26 0PZ, UK; claire.paton@moredun.ac.uk (C.P.); ross.bacchetti@moredun.ac.uk (R.B.); hannah.shaw@moredun.ac.uk (H.S.); williestew3383@aol.com (W.S.); frank.katzer@moredun.ac.uk (F.K.);; 2Scottish Natural Heritage, 54-56 Junction Road, Kirkwall, Orkney KW15 1AG, UK; James.Plowman@snh.co.uk

**Keywords:** *Cryptosporidium parvum*, calves, geese, catchments, public health

## Abstract

The parasite *Cryptosporidium*
*parvum* represents a threat to livestock health and production, water quality and public health. Cattle are known to be significant reservoirs of *C. parvum*, but transmission routes are complex and recent studies have implicated the potential role of wildlife in parasite transmission to cattle and water sources. On the Orkney Isles, high densities of Greylag geese (*Anser anser*) cause widespread faecal contamination of cattle pastures, where cryptosporidiosis is known to be the main cause of neonatal calf diarrhoea and *Cryptosporidium* contamination frequently occurs in two reservoirs supplying Mainland Orkney’s public water. This study aimed to determine the *Cryptosporidium* species and subtypes present in geese and calves co-grazing on four farms surrounding two reservoirs on Mainland Orkney. Results indicated a high level of *C. parvum* prevalence in calves, geese and water samples. gp60 analysis illustrated that higher genotypic diversity was present in the goose population compared with calves, but did not yield sequence results for any of the water samples. It can be concluded that the high levels of *C. parvum* evident in calves, geese and water samples tested represents a significant risk to water quality and public health.

## 1. Introduction

*Cryptosporidium parvum* is an environmentally ubiquitous parasite, responsible for causing the disease cryptosporidiosis in neonatal calves, as well as lambs, deer calves and humans, where it can cause particular problems in the young, elderly or immuno-compromised. Cryptosporidiosis is a gastro-intestinal disease for which profuse diarrhoea is the main clinical symptom, leading to rapid dehydration and potentially death in susceptible hosts [1]. Livestock, in particular, calves, are known to be the main reservoirs of *Cryptosporidium parvum,* a zoonotic species known to be responsible for 40% of human cryptosporidiosis cases in the UK [2]. Infected calves can shed billions of infective oocysts into the environment [3,4] but it has previously been shown that wildlife and other livestock, such as lambs, can contribute to environmental parasite loading. However, reports to date have been highly variable regarding prevalence and relative contribution of *C. parvum* from wildlife species [4,5,6,7,8,9,10].

The environmental stage of the parasite, the oocyst, is extremely tough and can survive for prolonged periods in favourable climatic conditions, such as damp and humid climates [11]. For these reasons, water is considered an important mechanism in the transmission of *Cryptosporidium* [12]. In addition, livestock pasture frequently surrounds catchment areas collecting water ultimately destined for human drinking water, which frequently causes problems for water providers relating to contamination with zoonotic pathogens. It is, therefore, critical to have accurate information on the prevalence of *Cryptosporidium* species present in catchments to assess the risk to public health from zoonotic transmission of *Cryptosporidium* through drinking water, and to understand parasite transmission dynamics more thoroughly. A better understanding of how the parasite behaves at a whole catchment level is critical [13].

Due to increasing contamination events of public water supplies with *Cryptosporidium*, the Scottish Water Directive (2003) was introduced to legislate for routine sampling of all public water supplies depending on *Cryptosporidium* risk. Risk assessments are calculated using weightings for parameters which affect *Cryptosporidium* levels for individual catchments or water supplies. One of the highest weightings is given to the presence of livestock in the catchment, where weighting score doubles if calves or lambs are present, or if grazing densities are high [14]. The risk weighting is increased if livestock have direct access to the water course and reduced if the livestock are fenced off from the water body. Wildlife are also considered to represent a zoonotic risk to water supplies but have a lower weighting than livestock, reflecting the generally lower grazing densities. This is not always the case, however, as wildlife populations in specific catchment areas can outnumber that of livestock (Orkney Goose Management Group; Pers. Comm.). In Mainland Orkney, through regulatory testing of reservoirs which are the source of the public water supplies, it is known that there is a high environmental loading of *Cryptosporidium* (Scottish Water; Pers. Comm.). This island is renowned for its high-quality beef production, which is the main livestock industry on Orkney, with spring calving being commonly carried out indoors during March, April and May with calves being turned out on to pasture as soon as weather permits, but generally, during May. Reports from local veterinary surgeons have confirmed that cryptosporidiosis is one of the commonest causes of neonatal calf scour in Orkney, which is reflected in the statistics for the UK (Veterinary Investigation Diagnostic Analysis (VIDA) Reports 2016–2018). Cryptosporidiosis, caused by infection with *C. parvum,* is a serious issue for livestock farmers as it significantly affects calf growth, production and suckler herd efficiency (H. Shaw; manuscript in preparation) and is proving very difficult to control on Orkney beef farms, despite rigorous management efforts from the farmers and vets concerned (NorthVets, Kirkwall, Orkney; Pers. Comm.).

Resident and migratory geese, which co-graze in high numbers with young calves on pasture and move freely from field to field, farm to farm, and in the case of migratory geese, between countries, have been suggested as a possible transmission vehicle for *C. parvum* (Orkney Goose Management Group; Pers. Comm.). There is very little published information on the role of geese in the transmission of zoonotic pathogens to livestock or humans, but some previous catchment studies have indicated that geese may act as potential vectors for *C. parvum* [15,16,17]. It has also been suggested that the high faecal loading of pathogens in geese may contribute to a significant risk of infection to other susceptible hosts [17]. In contrast, a recent comprehensive review focusing on a One Health perspective concluded that, based on present knowledge, there was not enough information to say whether geese played a role in the transmission of *Cryptosporidium,* but that previous research had potentially overrated the role of geese as disease vectors [18]. It should be noted that goose grazing densities were not included in any of the studies quoted in this review and it is accepted that the recorded numbers of both migratory and resident Greylag geese on Orkney are extremely high (Table 1 and Table 2) and that faecal contamination by the geese is widespread and occurs throughout the year (Orkney Goose Management Group; Pers. Comm.). Average counts for Greylag geese on Orkney are 22,025 (resident population over five years (2012–2016)) and 61,685 (resident and migratory populations over seven years (2012–2018)), reflecting the extremely high grazing densities of geese over an average land area of 101,735 hectares.

The high densities of geese have become a serious issue for farmers due to over-grazing of pastures and destruction of cereal crops, as well as large-scale faecal contamination of pasture. Greylag geese are a protected species, but as their numbers have increased to such an extent that they have become a problem for farmers, a goose management group has been established on Orkney, comprising representatives from Scottish Natural Heritage, the National Farmers Union of Scotland, local farmers, RSPB and Orkney Islands Council, to implement control strategies, including controlled culls of adult birds and oiling of eggs to prevent hatching. Despite this, although winter counts decreased over the time period 2012–2016, they have increased in the 2017 and 2018 winter counts, and summer counts have increased from 2013 to 2016 (Table 1 and Table 2) indicating the scale of the problem.

The aim of this pilot study was to determine the *Cryptosporidium* species and subtypes present in geese and calves co-grazing on four farms on Mainland Orkney surrounding two reservoirs supplying Mainland Orkney with water, and to analyse water samples, collected by Scottish Water for routine sampling, to establish potential transmission routes on and between farms and to assess water contamination levels and thereby risk to public health.

## 2. Materials and Methods

### 2.1. Sample Collection

Farms were selected on the basis of grazing proximity to Kirbister and Boardhouse reservoirs, which comprise Mainland Orkney’s public water sources, and where young calves were co-grazing with high densities of Greylag geese. Ethical approval was not required for this study and farmer permissions were obtained for each farm. Farm 1 grazed cattle with young calves in fields surrounding Kirbister reservoir where livestock had access to the reservoir in some unfenced areas and Farm 2 grazed cows and calves surrounding Boardhouse Loch. Farms 3 and 4 grazed cows and calves in fields surrounding Loch of Hundland, which drains directly into Boardhouse Loch. Freshly voided samples from calves and geese were collected from the ground in fields surrounding Kirbister, Boardhouse and Hundland lochs on mainland Orkney, on four farms identified by farmers as co-grazing young calves and Greylag geese during the period 15th May 2017 to 6th June 2017. Samples were stored in airtight containers with available quantities ranging from 50 g to 72 g for geese and 12 g to 36 g for calves. Calf ages ranged from one week old to six weeks old and all goose samples were collected from adult geese. Sampling was carried out with due care to avoid cross-contamination between geese and cattle samples, avoiding samples where they were within one metre of each other. Sample numbers collected from each farm are shown in Table 3. Collection of water from Kirbister (*n* = 26) and Boardhouse (*n* =20) reservoirs was performed by Scottish Water as part of regulatory sampling and according to Scottish Water’s standard operating protocols between the period March 2016 to February 2017 (http://standingcommitteeofanalysts.co.uk/Methods/Microbiology/drinkw.html).

### 2.2. Sample Processing and Analysis

#### 2.2.1. Processing Faecal Samples

Calf faecal samples: 250 μg of sample was added to 200 µL lysis buffer (T1 buffer, Macherey-Nagel, Duren, Germany. NZ740952250).Goose faecal samples: Salt flotation, using approximately 3 g of faecal sample, was performed [2], following which the final pellet was re-suspended in 200 µL lysis buffer. The extra salt flotation step was performed on goose samples due to the higher fibre content of these samples, which requires a further processing step prior to DNA extraction.

#### 2.2.2. DNA Extraction

All samples underwent 10 freeze–thaw cycles in liquid nitrogen and a water bath at 56 °C. DNA was extracted using NucleoSpin Tissue DNA, RNA and Protein Purification Kits (Macherey-Nagel, Duren, Germany. NZ740952250) following the manufacturer’s protocol with the following modifications: The samples were incubated with Proteinase K at 56 °C overnight, following which the samples were vortexed vigorously. Prior to the addition of ethanol, the samples were centrifuged at 11,000 × *g* for 5 min to remove insoluble particles and the supernatant was retained. Ultrapure water (100 μL) was used to elute DNA.

#### 2.2.3. PCR Sequencing and Analysis

Amplification of DNA was by nested PCR targeting the 18S gene [19]. Briefly, each 25 μL reaction contained 10 × PCR buffer (45 mM Tris–HCl pH 8.8, 11 mM (NH4)2SO_4_, 4.5 mM MgCl_2_, 4.4 μM EDTA, 113 μg mL^−1^ BSA, 1 mM each of four deoxyribonucleotide triphosphates), 0.5 units BioTaq (BIO-21040, Bioline, London, UK) and 10 μM of each primer. DNA (3 μL) was added in the primary round and 1 μL primary PCR product in the secondary round, after a 1:50 dilution with dH_2_O. The total volume was made up to 25 μL with dH_2_O. All reactions were carried out in triplicate and a positive DNA extraction and negative control (dH_2_O) were included on each plate. Cycling conditions were 3 min at 94 °C, followed by 35 cycles of 45 s at 94 °C, 45 s at 55 °C and 1 min at 72 °C. The final extension was 7 min at 72°C. Secondary amplification products (3 μL) were visualised on an AlphaImager 2000, following electrophoresis on a 1.5% Agarose gel stained with GelRed^TM^ (41002, Biotium, Fremont, CA, US).

All *Cryptosporidium*-positive samples were sent for Sanger sequencing (MWG Operon). The sequence results were aligned with reference 18S rRNA sequences (GenBank, NCBI) for each possible Cryptosporidium species using BioEdit software (Version 7.1, Informer Technologies Inc.) [20].

#### 2.2.4. Subtyping *C. parvum*-Positive Samples

For all *C. parvum*-positive samples, a region of the 60-KDaglycoprotein (gp60) gene was amplified and sequenced to assign gp60 subtype following a previously published protocol [21]. Briefly, a nested protocol was followed, amplifying a 450 bp region of the gene spanning the hypervariable polyserine tract in two rounds of PCR. Following this, PCR products were sequenced and aligned [21] and sub types named [22].

#### 2.2.5. Processing and Analysis of Water Samples

For water analysis, processing of filters, immunomagnetic separation (IMS) and microscopy were performed according to standard operating protocols (SOPs) by the Microbiology Laboratory, Scottish Water [23] Oocysts were identified microscopically using fluorescein isothiocyanate (FITC)–anti-*Cryptosporidium* monoclonal antibody (MAb) (FITC–C-MAb) and the nuclear fluorogen 4, 6-diamidino-2-phe-nylindole (DAPI) according to the Drinking Water Quality Regulator for Scotland (DWQRS) Standard Operating Protocol for Monitoring of *Cryptosporidium* Oocysts in Treated Water Supplies (http://www.dwqr.org.uk/technical/information-letters/public-2010). For each water sample collected and analysed by Scottish Water, one slide was produced. Slides with identified *Cryptosporidium* oocysts were collected from Scottish Water and the oocysts removed by adding 12 μL lysis buffer into the slide well and scraping the well with a loop. The liquid was then aspirated from the well into a tube containing 200 μL lysis buffer and the method followed as described for DNA extraction from calf and goose faecal samples with the additional step of two elutions using 50 μL ultrapure (UP) H2O followed by 25 μL UP H2O to maximise DNA yield. DNA amplification and subtyping of *C. parvum*-positive samples were as described for calf and goose samples.

## 3. Results

As a mean of all four farms, 48.7% (38/78) of calf samples and 26.0% (26/100) of geese samples were positive for *Cryptosporidium,* where 32.1% of the calf samples (25/78) and 24.0% of the goose samples (24/100) analysed were *C. parvum*-positive (Table 4). In calves, the majority of *Cryptosporidium*-positive samples were *C. parvum* (65.8%) and 40.0% of the *C. parvum*-infected animals had mixed infections with other *Cryptosporidium* species, whereas mixed infections were not detected in any of the geese. Of the geese samples positive for *Cryptosporidium*, the majority were *C. parvum* (92.3%) with only 3.8% *C. andersoni* and Goose subtypes. Results for the raw water samples from the two reservoirs (*n* = 46) showed that 73.9% of the total number analysed were *Cryptosporidium*-positive, with 44.1% of these positive samples being *C. parvum,* 52.9% *C. andersoni* and 2.9% *C. ubiquitum.* Therefore, a total of 47.0% of the *Cryptosporidium*-positive samples comprised zoonotic *Cryptosporidium* species or 34.8% of the total water samples analysed.

On the basis of individual farms, it is evident that there was variation between the prevalence and species of *Cryptosporidium* found in calf and goose samples (Figure 1 and Figure 2). For example, Farm 1 was the only farm where *C. andersoni* was isolated in both calves and geese, whereas calves on Farm 4 had a higher prevalence of *C. parvum* and *C. parvum* mixed infection.

The *Cryptosporidium* species prevalence found in water samples from the two reservoirs (Figure 2) showed a predominance of *C. parvum* and *C. andersoni*, reflecting the predominant species in calves and geese in these catchments.

On an individual catchment level, Farm 1 calves and geese were grazing the Kirbister catchment, and Farms 2, 3 and 4, the Boardhouse catchment. The relative prevalence of *C. parvum* in both catchments was very similar (Figure 2) and this is reflected by the *C. parvum* prevalence found in the geese and calves across the two catchments. Farm 1 had the highest prevalence of *C. andersoni* in calves particularly (Figure 1), which was also evident in the water samples from Kirbister catchment (Figure 2).

The *C. parvum*-positive samples from calves, geese and water underwent further analysis to determine gp60 subtypes to investigate *C. parvum* transmission. The predominant subtype found in calves on all four farms was IIaA15G2R1, with only one further subtype, IIaA15R1, detected. Figure 3 illustrates that geese showed more *C. parvum* genotypic diversity when compared with calves, which is evident on all four farms, with calves on both Farms 1 and 2 showing only one subtype, whereas there were three subtypes present in geese on Farm 1 and two on Farm 2. The calves on Farms 3 and 4 both had two subtypes present, whereas the geese on Farm 3 had four subtypes and two on Farm 4.

Unfortunately, despite repeated attempts to subtype the *C. parvum*-positive water samples, including concentrating the DNA and adapting the PCR protocol, no gp60 sequences were obtained.

## 4. Discussion

The prevalence of *Cryptosporidium*, and in particular, *C. parvum*, in calves was high (48.7% *Cryptosporidium*-positive with the majority being *C.-parvum* positive (65.8%)) and although comparable studies show a wide range in *C. parvum* infection rates, it is accepted that cryptosporidiosis is endemic in cattle worldwide [10,24,25,26]. *Cryptosporidium* prevalence figures are dependent on many factors, including the age of the calves at date of sampling, so even in studies using similar detection methods, high variation can be evident. *Cryptosporidium* prevalence in the geese samples analysed was lower compared to the calves, with 26.0% *Cryptosporidium*-positive samples and 24.0% *C. parvum*-positive samples. However, *C. parvum* was detected in 92.3% of the *Cryptosporidium*-positive samples from geese, suggesting either that geese are more susceptible to this species, or that this is a reflection of high *C. parvum* environmental contamination. It is interesting to note that the water samples over the two reservoirs showed a higher prevalence of *C. andersoni* when compared to *C. parvum*, suggesting environmental contamination was high for both *Cryptosporidium* species and, therefore, that geese are more susceptible to *C. parvum*. This is a very important finding when considering the epidemiology of *C. parvum*, an environmentally ubiquitous, zoonotic species of *Cryptosporidium*. Geese are highly mobile birds with the ability to move freely between farms, regions and, sometimes, countries and as this study suggests, they are susceptible to *C. parvum*, and they may be considered as important vectors and a risk to calves, humans and water contamination.

The *Cryptosporidium* species present, particularly in calves, varied across the farms and reflected the different age groups of the calves at the time of sampling. For example, the calves on Farm 1, where *C. andersoni* was prevalent, were older (4–6 weeks old at time of sampling) than on any of the other farms sampled, which is consistent with *C. andersoni* being more frequently detected in adult cattle and older calves [4]. In contrast, Farm 4 was calving later and had only 12 very young calves available at time of sampling, which had recently been turned out. This may be a reason why there was a lower prevalence of *C. parvum* found in the geese samples on Farm 4 (see Figure 1).

There is little published information on the prevalence of *Cryptosporidium* in wild geese, but from the available studies involving wild birds, a *Cryptosporidium* prevalence of 5.8% in wild aquatic birds including Greylag geese [27]; 2.4% in wild birds [28] and 5% in wild gulls [29] has been reported. The geese sampled in the present study were all wild birds and, as the sampling period was in the summer, comprised resident geese only. This could explain the high prevalence of *Cryptosporidium* in the current study, as resident geese would be grazing throughout the seasons with neonatal livestock, as well as juvenile and adult animals. Some research has suggested that the role of geese in the transmission of *C. parvum* appears to be limited to the geese acting as vectors without showing clinical signs of infection [30,31,32]. However, it has been shown that *C. parvum* oocysts retain their infectivity and viability after intestinal passage in Canada geese (*Branta canadensis*), with serious epidemiological implications [33]. Water-fowl can serve as mechanical vectors for water-borne oocysts and can contaminate surface waters with *C. parvum*; therefore, it is likely that even if *C. parvum* transmission is solely by mechanical transfer in geese, they are capable of transmitting viable *C. parvum* oocysts on to pasture, as well as water sources, and should, therefore, be considered as a risk to livestock, water quality and public health. The latter point being particularly important if grazings are near public water supply sources, such as reservoirs. This agrees with the findings of a US-based study investigating the role of geese and deer in a suburban/urban watershed, which detected *C. parvum* as well as *C. hominis*-like subtypes in geese and in the local watercourses. The authors concluded that these animals should be considered as vectors of human infectious *Cryptosporidium* species and, as such, should be targets for source water protection [15]. In the current study, as there was no opportunity to perform histopathology on any of the geese, it cannot be concluded if the geese were acting as *Cryptosporidium* vectors or were infected with the parasite. This information would be very valuable to obtain in a future study.

The results of the water sample analysis in the current study illustrated very high environmental *Cryptosporidium* contamination in both reservoirs (73.9% of water samples (*n* = 46) were *Cryptosporidium* positive) with zoonotic species being detected in 47.0% of these positive samples. Interestingly, *C. andersoni* was detected in higher prevalence in the samples from Kirbister reservoir and also, in Farm 1 calves (and one goose) grazing the surrounding fields, providing further evidence of *Cryptosporidium* transmission from grazing animals to water. The species of *Cryptosporidium* isolated from water sampling sites has previously been found to reflect the predominant species found in the livestock and wild deer at that particular time [10], providing further evidence for direct transfer of oocysts from grazing animals into the catchment water systems. This has also previously been recorded in surface water contamination with *C. parvum*, which was linked to calves grazing near the water course [34], a finding also confirmed by Robinson et al. [13] in a similar catchment-based study.

In a previous catchment study [10], gp60 subtyping of *C. parvum*-positive samples from livestock, deer and water suggested that transmission was occurring from both livestock species and deer into the water courses. Unfortunately, in the current study, no amplification at the gp60 locus was obtained from DNA extracted from water samples. This is likely to have been due to low parasite DNA concentrations and as gp60 is a single copy gene, this represents a disadvantage to current protocols, which have been optimised for animal samples, when applied to environmental samples with anticipated lower parasite numbers.

Subtypes of *C. parvum* were obtained for calf and goose samples using gp60 marker (Figure 3). The predominant subtype found in calves from all four farms was IIaA15G2R1, a subtype commonly found in calves [21] and often responsible for serious disease outbreaks on farms. The increased subtype diversity evident in the geese samples was interesting and potentially reflected the ability of the geese to move between farms and regions. In this respect, geese are likely to be important vectors of *C. parvum* strains, potentially moving these strains over long distances during migration, where they may be a source of infection for susceptible livestock and humans. The amount of time geese spend on water varies throughout the year, increasing when they have young and during the moult, and decreasing during the breeding season and for the rest of the year. At these peak times, the transmission of *C. parvum* and unusual strains, in particular, to public water supplies may be important for public health. The reservoirs on Orkney are shallow in depth, high in sediment loading and subject periodically to high wind turbulence. This results in high turbidity in the water and pressure on the filtration membranes in the treatment plant, causing breakthrough into the drinking water, which is a concern for the water industry (Scottish Water; Pers. Comm.). It has been suggested that much of the sediment is derived from goose faeces, as it is high in urea, and this will be investigated in a future study.

## 5. Conclusions

In Orkney, very high densities of resident and migrant Greylag geese co-graze with cattle in pastures surrounding Kirbister and Boardhouse reservoirs, which are the sources of Mainland Orkney’s public water supply. The extremely high goose numbers involved and unexpectedly high prevalence of *C. parvum* found in these geese may be a risk factor in the transmission of *C. parvum* to the water courses, where faecal pollution and *Cryptosporidium* contamination is an on-going issue. The results of the raw water analysis from both reservoirs emphasised the extent of the contamination in these water bodies. As part of water quality management strategies, fencing livestock off from the reservoir edges is ongoing and will be an important strategy for *Cryptosporidium* reduction in reservoirs. However, the results from this pilot study have suggested that management strategies designed to improve water quality will need to take the potential contamination from geese in such high densities into account. As this study sampled a relatively small number of animals, further research is planned in these catchments to improve the sample size of geese in particular, but also to sample water and sediment in the same time frame to ascertain if contamination hot-spots occur in areas of the reservoirs and if goose faeces is involved in sediment overloading.

## Figures and Tables

**Figure 1 microorganisms-07-00513-f001:**
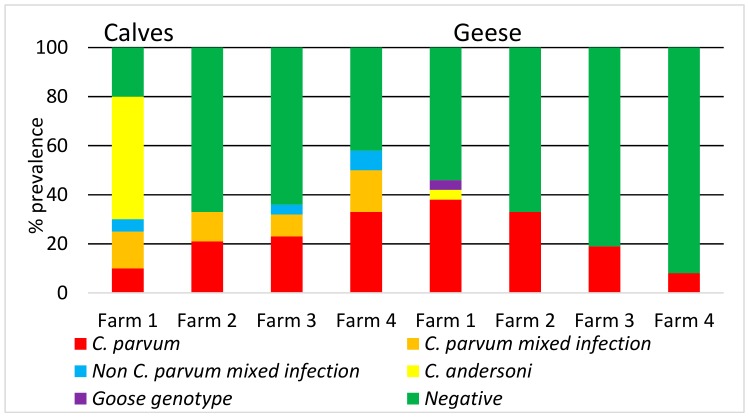
*Cryptosporidium* species prevalence (%) found on each farm in calves and geese.

**Figure 2 microorganisms-07-00513-f002:**
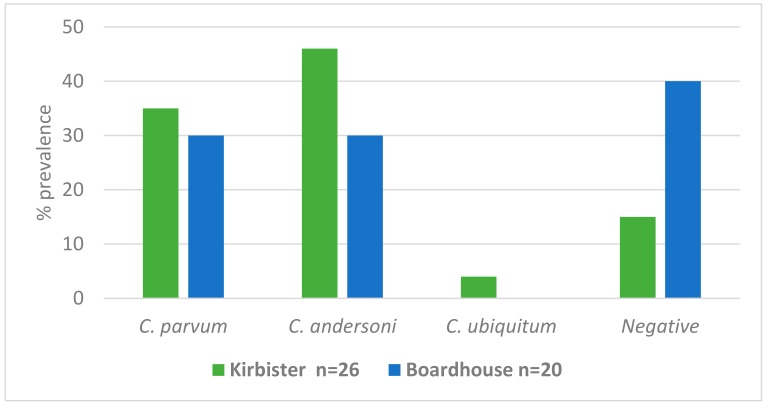
*Cryptosporidium* species prevalence (%) found in water samples from Kirbister and Boardhouse reservoirs.

**Figure 3 microorganisms-07-00513-f003:**
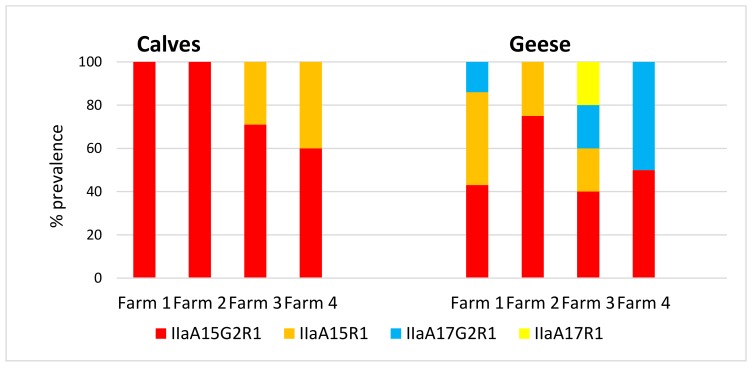
Percentage of *C. parvum*-positive samples with each identified genotype present in calf and goose samples.

**Table 1 microorganisms-07-00513-t001:** Orkney Greylag geese counts for 2012 to 2016 for summer counts and Table 2. 2012 to 2018 for winter counts (data supplied by Scottish Natural Heritage). Summer (August) counts—resident population.

Location	Area (ha)	Number—August Count					Mean Number over 5 Years
		2012	2013	2014	2015	2016	
North Ronaldsay	690	389	132	355	546	401	365
Sanday	5043	2591	1780	2083	2613	2578	2329
Westray	4713	840	1223	983	962	1082	967
Papa Westray	933	343		157	501	61	263
Eday	2745	1138	1221	708	566	814	889
Small Holms (Faray, Muckle Green Holm)	265	92	NC	NC	NC	NC	N/A
Stronsay	3430	951	1895	1978	1732	3477	2007
Shapinsay	2948	1765	1423	1282	1563	1915	1590
Rousay/Eynhallow	4935	399	113	576	447	568	419
Egilsay	650	0	36	146	176	20	76
Wyre	311	0	0	0	27	10	7
Gairsay	240	55	80	160	47	20	72
Auskerry	85	20	30	NC	25	12	22
East Mainland	52,325	2216	2233	1862	1952	2931	2239
West Mainland		8409	7660	9759	7012	7835	8135
Copinsay	73	0	0	NC	0	0	N/A
Burray	1098	731	750	571	466	352	574
South Ronaldsay	4980	1234	1370	2233	2113	2021	1794
Hoy & South Walls	14,558	107	271	0	495	0	175
Flotta / Fara / Switha	1212	87	25	58	6	142	64
Graemsay	409	0	0	0	105	11	23
Swona	92	NC	0	NC	NC	NC	N/A
**TOTAL**	**101,735**	**21,367**	**20,242**	**22,911**	**21,354**	**24,250**	**Mean 22,025**

**Table 2 microorganisms-07-00513-t002:** Winter (November) Count Totals—resident and migratory geese.

Year	Total Orkney
2018	63,534
2017	63,045
2016	46,678
2015	56,151
2014	65,067
2013	63,665
2012	74,913
**MEAN**	**61,865**

**Table 3 microorganisms-07-00513-t003:** Numbers of calf and goose samples collected from each farm.

Farm	Numbers of Calf Samples	Numbers of Goose Samples
1	20	24
2	24	24
3	22	26
4	12	26
Total	78	100

**Table 4 microorganisms-07-00513-t004:** *Cryptosporidium* species found in calf and goose samples on four Orkney farms and in water samples from two reservoirs.

*Cryptosporidium* Species	Identified Cryptosporidium Species Calves Stool Samples N (%)	Geese Stool Samples N (%)	*** Water Samples N (%)
*C. parvum*	19 (15/78)	24 (24/100)	33 (15/46)
** C. parvum* mixed infection	13 (10/78)	0	0
** Non *C. parvum* mixed infection	6 (5/78)	0	0
*C. andersoni*	9 (7/78)	1 (1/100)	39 18/46)
*C. bovis*	1 (1/78)	0	0
*C. ubiquitum*	0	0	2 (1/46)
Goose genotype	0	1 (1/100)	0
**Total No.**	**78**	**100**	**46**

* *C. parvum* mixed infections included *C. parvum*, *C. andersoni* and *C. bovis*. ** Non *C. parvum* mixed species comprised *C. andersoni*; *C. bovis* and *C. ryanae*. *** Water sample is one slide equivalent.

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
