# Peer review of "Cryptosporidium Prevalence in Calves and Geese Co-Grazing on Four Livestock Farms Surrounding Two Reservoirs Supplying Public Water to Mainland Orkney, Scotland"

_microorganisms, 2019, doi:10.3390/microorganisms7110513_

Round 1

Reviewer 1 Report

In the article entitled “Cryptosporidium prevalence in calves and geese cograzing on four livestock farms surrounding two reservoirs supplying public water to Mainland Orkney” authors describe the prevalence of zoonotic species of Cryptosporidium in cattle, birds and water. Studies on this topic are scarce and represent a contribution to our knowledge about Cryptoporidium transmission. However some aspects have to be clarified and some modifications concerning the presentation are required.

General comments:

Material and methods:

Concerning methodology I have several questions:

An ethical approval was not necessary for carrying on this study?

It would be useful to have a map showing the different sampling areas.

Authors should give more details in the methodology: how samples were collected, amount of collected stools, age of animals for cattle, conservation. Concerning processing, why treatment was different for calf compare to goose fecal samples?

Were sequences deposed in Genbank?

Results

I understand that samples were directly collected from the ground. A cross contamination between birds and cattle fecal samples was not possible once the samples were in the field? It is no clear if authors considered geese as infected or as only carriers of Cryptosporidium.

Can you precise the ratio for 48.7% and 26% (page 5, line 182)

Table 3 and Figure 1 are redundant. Figure 1 can be eliminated.

In table 3 is not clear how authors define a water sample, by number of screened slides? Can you precise in material and methods, and also with a footnote in the table?

Table 3 has to be reorganized:

In the first column, first line: Replace “species” by “Cryptosporidium species”

Add a heading for calves, geese and water indicating : “Identified Cryptosporidium species”. Bellow the heading modify replacing by: “Calves stool samples N (%)”, “Geese stools samples N (%)”, “Water samples N (%)”.

Delete “only” everywhere.

Line 9 corresponding to negative can be deleted.

Discussion:

Authors should include more discussion about the fact that they found a zoonotic species of Cryptosporidium in geese samples.

Minor comments,

Title: Add “Scotland” at the end.

Add 60-KDaglycoprotein for the first time that gp60 is mentioned.

Replace GP60 by gp60 through the manuscript.

Page 2, line 55 and page 5 line 166: Put a number and add the references in the bibliography.

Page 3, Line 94: This table does not have a number.

Page 4, line 140: Modify the title as follows: “PCR, sequencing an analysis”.

When referring to GP60 it is better to refer to subtyping and not to genotyping. This has to be changed through the manuscript.

Page 5, line 155: Cryptosporidium should be in italics.

Page 5, line 162-163: The first sentence can be deleted, it is a repetition: “Water samples…for Cryptosporidium”. Add: “For water analysis” at the beginning of the next sentence.

Table 3: The details of the species can be placed in a footnote.

Figure 2: Add “infection” after “non C. parvum mixed”

Page 7, line 206: Modify as follows: “The Cryptosporidium species prevalence found in water samples from the two reservoirs”…

Figure 4: The name of the y axe is missing.

Page 8, line 238-239: Modify as follows: “with 26% of Cryptosporidium positive samples and 24% of C. parvum positive samples…”

Page 8, line 289: Replace “were” by” was”.

Page 9, line 261: replace “which has” by” with”.

Page 9, line 270: Modify as follows: “human infectious Cryptosporidium species”.

Page 9, line 292: Replace “strain” by “subtype”.

Page 9, line 294: Replace “genotypic” by “subtype”.

Page 10, line 311: modify as follows: “may be a risk factor”.

Reviewer 2 Report

Your research is of high interest because only few attention has been done to birds as possible vectors of parasitic infections until now.

This is a factor to be better investigated inside the zoonotic transmission of pathogen protozoa. 

Author Response

Note to reviewer 2: Thank you for your positive comments

Round 2

Reviewer 1 Report

The manuscript has improved compared to the previous version but some modifications are still needed. I have the following suggestions for the authors to consider:

In Table 3 the percentage have to be included.  

It is possible to delete the line with negative results if numbers are presented as follows:Taking the first line as example:

C. parvum, 15/78 (19.23), 24/100 (24), 15/46 (32.6).

Page 1, line 20: Replace GP60 by gp60.

Page 2, line 55: Keep only the number of the reference.

Page 4, line 137: Can you detail the amount of goose sample that was used ? Try to present only one pargraph.

Page 5, lines 175-178: Keep only the number of the reference

Figure 4 : The axe name is still missing. Maybe a problem during PDF assemblage.

References :

Italics are missing when naming genus/species.

Check the bibliography. There is a problem with some references, for instance : Reference 2.

Author Response

Thank you for reviewing our manuscript and for your valuable comments.
